# QUANTUM SEQUENTIAL SCATTERING MODEL FOR QUANTUM STATE LEARNING

## ABSTRACT

Learning probability distribution is an essential framework in classical learning theory. As a counterpart, quantum state learning has spurred the exploration of quantum machine learning theory. However, as dimensionality increases, learning a high-dimensional unknown quantum state via conventional quantum neural network approaches remains challenging due to trainability issues. In this work, we devise the quantum sequential scattering model (QSSM), inspired by the classical diffusion model, to overcome this scalability issue. Training of our model could effectively circumvent the vanishing gradient problem to a large class of high-dimensional target states possessing polynomial-scaled Schmidt ranks. Theoretical analysis and numerical experiments provide evidence for our model's effectiveness in learning both physical and algorithmic meaningful quantum states and show an out-performance beating the conventional approaches in training speed and learning accuracy. Our work has indicated that an increasing entanglement, a property of quantum states, in the target states, necessitates a larger scaled model, which could reduce our model's learning performance and efficiency.

## 1 INTRODUCTION

The innovation of classical machine learning has brought significant convenience and efficiency in industry and society. In particular, learning distributions between individual events and data is one of the crucial tasks for multiple usages in decades Anderson et al. (1977); Geng (2016). A plethora of approaches and schemes have been designed to learn probability distributions, such as continuous evolutionary algorithms Hansen et al. (2015); Kern et al. (2004) and supervised learning within the neural network (NN) framework including Boltzmann machine, graph neural network and diffusion model Baum & Wilczek (1987); Franceschi et al. (2019); Hoogeboom et al. (2021)

Meanwhile, by the fast growth of the requirement on computational power, quantum computing, as a prospective new framework, is expected to provide advantages over classical technology. The remarkable achievements from classical machine learning models LeCun et al. (2015); Serban et al. (2016) have spurred the generation of their counterparts within the field of quantum machine learning (QML) Biamonte et al. (2017); Schuld et al. (2015); Lloyd et al. (2013); Schuld et al. (2014). Quantum neural networks (QNNs) composed of layers of parametrised quantum circuits have received massive attention regarding various architectures addressing computation challenges Rebentrost et al. (2018); Zhao et al. (2019); Cong et al. (2019), including quantum state learning.

In quantum, the correlations between quantum data are encoded in the quantum states. Consequently, the task of learning an arbitrary quantum state bears a resemblance to classical distribution learning, which has inspired developments of state learning QML models Chowdhury et al. (2020); Ghosh et al. (2019); Wang et al. (2021a). As a main solution to quantum state learning, however, the implementation of the QNN-based methods suffers obstacles in efficiency, scalability and trainability. Specifically, training deep QNNs composed of multiple layers can experience exponentially vanishing gradients, or called *barren plateaus* (BP) McClean et al. (2018) when targeting high-dimensional states.

This work proposed a quantum sequential scattering model (QSSM) to overcome this bottleneck in QNN-powered state learning techniques. We provide both theoretical and numerical demonstrations of QSSM on training efficiency and learning accuracy, which can outperform the conventional QNN

model using universal layers. Recent research on the trainability issue of QNNs indicates prospective directions by reducing the expressibility of QNN architectures Cerezo et al. (2021); Liu et al. (2022a), adopting clever parameterization strategies Grant et al. (2019); Kulshrestha & Safro (2022); Volkoff & Coles (2021); Friedrich & Maziero (2022) and using adaptive algorithms Grimsley et al. (2019); Zhang et al. (2021); Skolik et al. (2021); Grimsley et al. (2022).

We drew inspiration from the classical diffusion model Yang et al. (2022) by conducting the state learning with progressively augmenting sublevels in a sequential manner. Our model combines the ideas of quantum purification theory and adaptive and layerwise training Quek et al. (2021); Skolik et al. (2021) for which the training process can be treated as the dilation of quantum information from subsystems to the entire one. The structure of the model ensures a dramatic reduction in the number of optimized parameters at each training step and, therefore, avoids barren plateaus for a large class of target states.

## 2 PRELIMINARIES

### 2.1 CLASSICAL DISTRIBUTION LEARNING

We briefly introduce the formalism concerning classical probability distribution learning. Correlations between discrete data variables, denoted as $X$, can be characterized by some probability distributions $D$ Kearns et al. (1994). The learning of such a distribution can be described as constructing a generator $G_{D'}$ that takes $x \in X$ as an argument and outputs $G_{D'}[x] \in X$ with respect to a distribution $D'$. The generator can be realized via a classical machine learning model, which is trained to achieve $d(D, D') \leq \varepsilon$ for some legal metric $d$, e.g., *Kullback-Leibler divergence* Csiszar (1975), and some threshold error $\varepsilon$.

### 2.2 QUANTUM STATE LEARNING

A typical quantum state learning task for an unexplored target state $\rho$, as a density matrix, solves for a generator that can be efficiently constructed to produce a representation $\rho'$ which $\mathcal{D}(\rho, \rho') \leq \varepsilon$ resembling classical distribution learning. Here $\mathcal{D}$ is a feasible distance measure on matrix space. Such a generator can veritably produce $\rho'$ instead of numerically simulating it Vidal (2003) and can be repeatedly used in further computational tasks. This work focuses on the QNN-powered algorithms combining both classical and quantum computation. Utilizing parameterized quantum circuits working as the state generators that are trained by gradient descent or gradient-free methods to determine the optimal parameters Peruzzo et al. (2014); Kandala et al. (2017). Beyond our scope, schemes using shadow tomography Aaronson (2018); Huang (2022) fulfil another category of state learning with the aim of characterizing the classical information of quantum states.

### 2.3 QUANTUM COMPUTING & QNN LAYERS

Our notations follow the conventional textbook by Nielsen and Chuang Nielsen & Chuang (2010). For more, we invite readers to have access to the supplementary material Appendix A for more details on quantum computing and quantum machine learning.

Quantum information is encoded and processed via the fundamental cells, namely, qubits. An $n$-qubit state can be mathematically represented by a $2^n \times 2^n$ positive semi-definite density matrix $\rho$, i.e., $\rho \succeq 0$ over the complex field and $\text{Tr}[\rho] = 1$. A pure state, in this formulation, satisfy $\text{Rank}(\rho) = 1$ and can be expressed in *Dirac bra-ket* notation as $\rho = |\psi\rangle\langle\psi|$ where $|\psi\rangle \in \mathbb{C}^{2^n}$ denotes a *Hilbert space* unit column vector with the corresponding *dual vector* $\langle\psi|^\dagger = |\psi\rangle$ and $\dagger$ denoting the complex conjugate transpose operation. A mixed state satisfies $\text{Rank}(\rho) > 1$, and based on *Spectral theorem*, it has a decomposition form $\rho = \sum_j p_j |\psi_j\rangle\langle\psi_j|$ where $p_j > 0$ denotes the probability of observing $|\psi_j\rangle\langle\psi_j|$ in $\rho$ and $\sum_j p_j = 1$.

The evolution of a quantum state $\rho$ is realized by applying a series of quantum gates which are mathematically described as unitary operators. The state $\rho'$ that undergoes transformation via a quantum gate $U$ can be obtained through direct matrix multiplication, expressed as $\rho' = U\rho U^\dagger$. Common single-qubit gates include the Pauli rotations $\{R_P(\theta) = e^{-i\frac{\theta}{2}P} | P \in \{X, Y, Z\}\}$, which

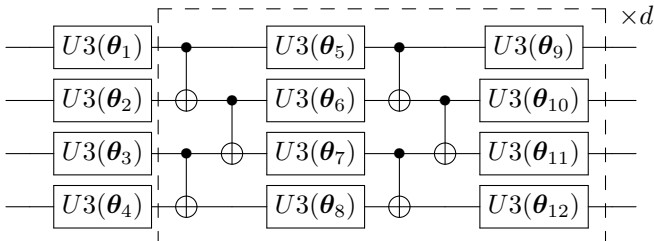

Fig 1: The general architecture of the QNN layers used for quantum state learning. The U3-gates can be decomposed as a combination of $R_Z(\phi_1)R_X(-\pi/2)R_Z(\theta_1)R_X(\pi/2)R_Z(\lambda_1)$ where the parameter vector $\boldsymbol{\theta}_1 = (\theta_1, \phi_1, \lambda_1)$. The layer consists of CNOT gates and U3 gates. The dashed block circuit repeats $d$ times as the depth of the layer. The above has a layer width $w = 4$, which applies to 4 quantum registers. In reality, the above circuit diagram represents a way of applying quantum gates sequentially in order from left to right.

are in the matrix exponential form of Pauli matrices

$$X := \begin{pmatrix} 0 & 1 \\ 1 & 0 \end{pmatrix}, Y := \begin{pmatrix} 0 & -i \\ i & 0 \end{pmatrix}, Z := \begin{pmatrix} 1 & 0 \\ 0 & -1 \end{pmatrix}. \tag{1}$$

Multi-qubit gates, e.g., controlled-$X$ gate CX (or CNOT) $= I \oplus X$ and controlled-$Z$ gate CZ$= I \oplus Z$ where '$\oplus$' denotes the direct sum operation live in high-dimensional linear operator space over $\mathbb{C}$. Quantum measurements working as projections are applied at the end of the quantum circuits. Quantum neural networks are usually formed by layers of parameterized circuits shown in Fig. 1 consisting of a bunch of single-qubit gates and several two-qubit gates.

## 3 MAIN RESULTS

In this paper, we design a quantum sequential scattering model (QSSM) absorbing the ideas of classical diffusion model and adaptive learning Quek et al. (2021), which has modular structured parametrised circuits, or we called the *scattering layer*, at each training step. Each layer ensures learning the reduced density matrix of a specific part in the target state so that the model can gradually rebuild the entire state after accomplishing all training steps.

Our main contributions involve **(1)** conceptually proposing the idea of combining quantum information diffusion and adaptive quantum state learning, **(2)** technically devising a new quantum neural network model, namely QSSM and the state learning algorithm via a sequentially subsystem-learning strategy, **(3)** theoretically proving the effectiveness of the state learning algorithm and a polynomial-scaled gradient variance of QSSM which indicates an avoidance of barren plateaus for rank-restricted state learning, **(4)** numerically demonstrating our results on learning different quantum states involving the noise effects. We compare QSSM directly to the conventional QNN model for handling state learning tasks and showcase its enhancement in both training efficiency and learning accuracy. The main results are presented in the following sections.

### 3.1 QUANTUM SEQUENTIAL STATE COMPOSITING

Quantum states are represented in a multiple-qubit system with a fixed order. We treat each qubit as a quantum register, just like classical bit and classical register, and label it $q_k$ for the $k$-th register. We then define a special characteristic for quantum states.

**Definition 1** *Given an $n$-qubit quantum state $\rho$ represented by $n$ ordered quantum registers labeled as $q_1, q_2, \cdots, q_n$, denoting $\rho_k$ as the $k$-th reduced density matrix of the first $k$-register state, i.e., $\rho_k = \mathrm{Tr}_{q_{k+1}:q_n}[\rho]$ for $1 \le k \le n$ where the operation $\mathrm{Tr}_{q_i:q_j}[\cdot]$ representing a partial tracing over registers $q_i$ to $q_j$, the (Schmidt) rank sequence of $\rho$ is an ordered list $\mathcal{R}_\rho$,*

$$\mathcal{R}_\rho = \{r_1, r_2, \cdots, r_{n-1}, r_n\}, \tag{2}$$

*where $r_k$ indicates* $\mathrm{Rank}[\rho_k]$. *In particular, if $\rho$ is pure, then $r_n = 1$ since $\rho$ can be represented as $|\phi\rangle\langle\phi|$ for some pure state vector $|\phi\rangle$.*

With these clarified, we could then present our sufficient and necessary conditions for QSSM to completely learn a target state using Algorithm 1, provided enough training time and layer width. Our analysis will concentrate on the pure target state $\rho$. However, the statement applies to the cases of mixed target states i.e., $\mathrm{Rank}[\rho] > 1$, since we could equivalently learn its purification state by introducing auxiliary systems. The formal version of proposition 1 can be found in Appendix B.

**Proposition 1** *For a given $n$-qubit pure target state $\rho$ represented by $n$ ordered quantum registers $q_1, q_2, \cdots, q_n$, if the rank sequence of $\rho$ is $\mathcal{R}_\rho = \{r_1, r_2, \cdots r_{n-1}, r_n\}$. Then there exists a quantum algorithm 1, based on QSSM, that could produce a state $\sigma$ exactly satisfying $\sigma = \rho$, if and only if the $k$-th scattering layer $U_k(\boldsymbol{\theta}_k)$ of QSSM has a width $w_k$ scales $\mathcal{O}(\lceil \log_2 r_k \rceil)$.*

We see that the width of each scattering layer scales only logarithmic regarding the target states' rank sequence. In general, even the rank of quantum pure state scales $\mathcal{O}(2^{\lceil n/2 \rceil})$, the logarithmic scaling in $w_k$ still guarantees a linear growth in the requirement of layer width concerning the number of qubits $n$, in the worst case.

Moreover, though many quantum states have full rank, there is a polynomial number of dominant components in their spectral decomposition. Learning their low-rank approximation pre-determined by the quantum principal component analysis (QPCA) Lloyd et al. (2014) can be treated as a quantum compressing of unknown states, which still captures the main statistical behaviours of target states. With a certain error tolerance for the low-rank approximation, the layer width can be further reduced, leading to more advantages in QSSM state learning. In the Numerical Simulations (Section 5), we provide evidence of learning different states' rank-restricted approximation.

Compared to the $n$-qubit universal-QNN model state learning, QSSM demands significantly fewer parametric degrees of freedom (DOF) to reach the same approximating error. The generating Lie algebra of an $n$-qubit universal QNN model has to span $\mathrm{SU}(2^n)$, resulting in a model DOF of $\mathcal{O}(4^n)$. On the contrary, since the $k$-th scattering layer involves at most $(\lfloor \frac{n}{2} \rfloor + 1)$ quantum registers, the total DOF of QSSM experiences a quadratic reduction to at most $\mathcal{O}(4^{\lfloor \frac{n}{2} \rfloor})$. Also, to learn the polynomial rank-bounded target state $\rho$, i.e., $r_{\max} = \max \mathcal{R}_\rho \sim \mathcal{O}(\mathrm{Poly}(n))$. The DOF required for each scattering layer in QSSM scales $\mathcal{O}(\mathrm{Poly}(n))$. Therefore, the entire model comprises fewer quantum gates, rendering this approach considerably more hardware-efficient.

### 3.2 AVOIDING BARREN PLATEAUS

Trainability is a critical challenge for the usage of quantum neural networks. Using a global deep QNN model brings stronger expressibility despite significantly increasing the randomness of initialization. Therefore, the initial gradient of trainable parameters in the model would exponentially vanish as the system scales up, called the Barren Plateau (BP) issue McClean et al. (2018).

With the diffusion of local quantum state information, QSSM has illustrated a potential to address trainability issues by focusing on subsystems in each scattering layer instead of the whole state. From the perspective of adaptive learning, we align the reduced quantum states of the $k$-th subsystem by minimizing the $k$-th adaptive cost function of equation 3 during the respective layer training,

$$C_k(\boldsymbol{\theta}) = \|\sigma_k(\boldsymbol{\theta}) - \rho_k\|_2^2 = \mathrm{Tr}\left[(\sigma_k(\boldsymbol{\theta}) - \rho_k)(\sigma_k(\boldsymbol{\theta}) - \rho_k)^\dagger\right], \tag{3}$$

where $\|A\|_2$ for some linear operator $A$ denotes the Schatten-2 norm, $\sigma_k(\boldsymbol{\theta}_k)$ and $\rho_k$ represent the $k$-th scattering layer produced state and the $k$-th reduced target state, respectively.

In this section, we show that QSSM has explicit advantages in trainability by investigating the statistical properties of the partial gradient with respect to particular layer parameters. For the cost gradient $\partial_\mu C_k$ regarding the $\mu$-th trainable parameter in the $k$-th scattering layer denoted as $U_k(\boldsymbol{\theta}) = U_+^{(k)}(\boldsymbol{\theta}_+)e^{-i\theta_\mu H_\mu}U_-^{(k)}(\boldsymbol{\theta}_-)$, all the parameters in the layer are represented in a parameter vector $\boldsymbol{\theta} = (\boldsymbol{\theta}_+, \theta_\mu, \boldsymbol{\theta}_-)$, where $\boldsymbol{\theta}_-$ and $\boldsymbol{\theta}_+$ represent the parameters of the forward and the backward parts within the $k$-th scattering layer having $e^{-i\theta_\mu H_\mu}$ centralized. The results are summarized.

**Proposition 2** *Given the state learning algorithm stated in Proposition 1, for an $n$-qubit pure target state $\rho$ represented by $n$ ordered quantum registers $q_1, q_2, \cdots, q_n$ with a rank sequence $\mathcal{R}_\rho = \{r_1, r_2, \cdots r_{n-1}, r_n\}$, if one of the $U_\pm^{(k)}$ in the $k$-th scattering layer $U_k$ forms at least local unitary 4-design, the expectation and the variance of $C_k$ with respect to $\theta_\mu$ can be upper bounded by,*

$$\mathbb{E}[\partial_\mu C_k] = 0; \quad \mathrm{Var}[\partial_\mu C_k] \in \mathcal{O}\left(\frac{g(\rho_k)}{r_k}\right), \tag{4}$$

*where the expectation is computed regarding the Haar measure and the factor $g(\rho_k)$ scales polynomially in $\mathrm{Tr}[\rho_k^2]$ known as the purity of $\rho_k$.*

The formal statement of Proposition 2 is presented in Appendix C. This proposition notably implies that the gradient magnitude is significantly determined by $r_{\max}$ in $\mathcal{R}_\rho$ rather than the total number of quantum registers $n$. In other words, the gradient magnitude can escape from barren plateaus by carefully setting the width of each scattering layer to adapt to the target state. A typical example is to learn an $n$-qubit GHZ state, which, by its symmetry, requires setting $w_k \leq 2$ for all scattering layers in QSSM and hence achieves $\mathcal{O}(1)$ upper bound in the variance of the gradient.

Moreover, Proposition 2 implies that QSSM can efficiently facilitate the learning of any pure states with polynomial-scaling $r_{\max}$ in $n$. This encompasses a broad class of quantum states, including *slightly entangled states* Vidal (2003) and *matrix product states* Perez-Garcia et al. (2006), which extends the efficient-learnable region of quantum states using quantum neural network models. Even in the case where $r_{\max}$ scales exponentially, the gradient magnitude still gains a square root enhancement by the bounded variance of $\mathcal{O}(2^{-\lfloor n/2 \rfloor})$ compared with the conventional model, scaling as $\mathcal{O}(2^{-n})$ to reach the same learning accuracy.

One may also apply the previous statement by allowing the error tolerance on the state learning and omitting the influence of the tail eigenvalues of the target states based on QPCA. Therefore, the efficient training condition of QSSM still applies to the low-rank state approximation learning by fixing a maximum scattering layer width.

## 4 QUANTUM SEQUENTIAL SCATTERING MODEL

The fundamental idea of state learning using the quantum sequential scattering model (QSSM) is to composite the target states by gradually aligning reduced density matrices of subsystems. The model diffuses the local quantum information into the global system, which can be considered a quantum analogy of the classical diffusion model. In contrast, the conventional QNN model handles the entire system at a time. We now present the overview of our QSSM with an efficient state learning algorithm.

Suppose we have access to the copies of an $n$-qubit pure target state $\rho = |\phi\rangle\langle\phi|$ from some other quantum instances. The target state can be represented in a system containing $n$ ordered quantum registers. Recalling $\rho_k$ as the reduced density matrix on the first $k$ registers, i.e., $\rho_k = \mathrm{Tr}_{q_{k+1}:q_n}[\rho]$, our model aims to construct a purification $|\psi_k(\boldsymbol{\theta}_k)\rangle = U_k(\boldsymbol{\theta}_k)|\psi_{k-1}\rangle$ of $\rho_k$ at the $k$-th learning step ($1 \leq k \leq n$) by training the $k$-th scattering layer realized as a parameterised circuit $U_k(\boldsymbol{\theta}_k)$. Notice that the learning results from the previous step are naturally involved in the state $|\psi_{k-1}\rangle$ having all first $k$ registers aligned.

The training of each layer is based on minimizing some adaptive cost functions, which in this work, we use the modified distance function of form 3 where the $k$-th layer output state $\sigma_k(\boldsymbol{\theta}_k) = \mathrm{Tr}_{q_{k+1}:q_n}[|\psi_k(\boldsymbol{\theta}_k)\rangle\langle\psi_k(\boldsymbol{\theta}_k)|]$. By hierarchically training the scattering layers until all registers are aligned, we could then construct the entire target through our trained quantum sequential scattering model. We summarize our quantum state learning algorithm via QSSM in Algorithm 1.

### 4.1 COST FUNCTION EVALUATION

As a hybrid quantum-classical model, we declare some details of the realization of the model in the following. For the adaptive $k$-th step cost function defined in equation 3. By rearranging equation equation 3 as,

$$C_k(\boldsymbol{\theta}_k) = \mathrm{Tr}[\sigma_k^2(\boldsymbol{\theta}_k)] + \mathrm{Tr}[\rho_k^2] - 2\,\mathrm{Tr}[\sigma_k(\boldsymbol{\theta}_k)\rho_k], \tag{5}$$

---

**Algorithm 1** Quantum sequential scattering model for (pure) state learning

---

**Require:** Copies of the $n$-qubit target state $\rho = |\phi\rangle\langle\phi|$, Cost tolerance $\delta$.
**Ensure:** The entire model has $n$ quantum registers as $q_1, q_2, \cdots, q_n$, and are initialized to $|0\rangle^{\otimes n}$.
**Parameter**: All layer parameters are randomly initialized regarding Uniform distribution of $[0, 2\pi)$.
Set $k = 1$ and maximum layer width $w_{\max}$.

  1: Update scattering layer width $w_k = k + 1$, $|\psi_k\rangle = |0\rangle^{\otimes n}$.
  2: **while** $k \leq n$ **do**
  3:    **if** $k \leq \lfloor n/2 \rfloor$ **then**
  4:      $w_k = \min\{k + 1, w_{\max}\}$.
  5:    **else if** $k > \lfloor n/2 \rfloor$ **then**
  6:      $w_k = \min\{n - k + 1, w_{\max}\}$.
  7:    **end if**
  8:    Apply $U_k(\boldsymbol{\theta}_k)$ to the quantum registers indexing $q_k$ to $q_{k+w_k-1}$, i.e., $q_k : q_{k+w_k-1}$.
  9:    Minimize $C_k(\boldsymbol{\theta}_k)$ via running classical training algorithm based on the analytic cost function
        and gradient $\nabla_{\boldsymbol{\theta}_k} C_k$ evaluations. The minimization stops until the cost difference reaches $\delta$.
10:    $k = k + 1$.
11:    Update $|\psi_k\rangle = U_k(\boldsymbol{\theta}_k)|\psi_{k-1}\rangle$.
12: **end while**
13: Store all optimized $\boldsymbol{\theta}_1, \cdots, \boldsymbol{\theta}_n$ in classical memory.
14: **return** model reconstructed representation $|\psi_n\rangle = U_n \cdots U_1 |0\rangle^{\otimes n} \approx |\phi\rangle$.
**Output**: The trained QSSM as an approximate state generator $\mathbf{U} = U_n \cdots U_1$ of target $|\phi\rangle$.

---

which is *convex* according to Theorem 2.10 of Carlen (2009).We chose this cost form since it can be efficiently evaluated on quantum hardware. The high-order state overlap terms involving $\text{Tr}[\rho^2]$ and $\text{Tr}[\rho\sigma]$ can be evaluated via *swap test* Barenco et al. (1997), which have been experimentally demonstrated on real quantum devices Islam et al. (2015); Linke et al. (2018). The training of the $k$-th layer can be described as finding the $k$-th step optimal parameters $\boldsymbol{\theta}_k^{opt}$ so that $C_k(\boldsymbol{\theta}_k^{opt})$ is minimized to approximately zero. To implement that, classical gradient-based and gradient-free methods, such as ADAM and COBYLA Kingma & Ba (2014); Powell (1994), can either be used during optimizations. Other metrics can also be employed in training procedures, and we left this aspect open for future research.

### 4.2 ANALYTIC GRADIENT EVALUATION

Further, the analytical gradients of the cost function in equation 3 can be computed efficiently, making the gradient-based scheme a prospective candidate for the training processes. According to Schuld et al. (2018); Mitarai et al. (2018); Ostaszewski et al. (2019); Wang et al. (2021b). Suppose the $k$-th layer $U_k$ consists of the gates satisfying the *parameter-shift rule* Mitarai et al. (2018); Schuld et al. (2018) and contains $m$ trainable parameters. Each optimization iteration is driven by the estimations of cost gradient given by,

$$\nabla_{\boldsymbol{\theta}_k} C_k(\boldsymbol{\theta}_k) = \Big(\partial_1 C_k(\boldsymbol{\theta}_k), \cdots, \partial_m C_k(\boldsymbol{\theta}_k)\Big), \tag{6}$$

where $\partial_\mu := \frac{\partial}{\partial \theta_k^\mu}$ indicating the partial derivative with respect to a fixed $\theta_k^\mu$ in the $k$-th layer. In particular, we derive the analytic gradient of $C_k$ as follows,

$$\partial_\mu C_k^* = \langle G_k^* \rangle_{(\theta_k^\mu)^* + \frac{\pi}{2}} - \langle G_k^* \rangle_{(\theta_k^\mu)^* - \frac{\pi}{2}} \tag{7}$$

The symbol $*$ indicating the corresponding quantity evaluated at $\boldsymbol{\theta}_k = \boldsymbol{\theta}_k^*$. $G_k$ is a Hermitian operator involves both $\sigma_k$ and $\rho_k$ having an expression,

$$G_k(\boldsymbol{\theta}_k) := \Delta_k(\boldsymbol{\theta}_k) \otimes \Gamma_k \tag{8}$$

where $\Delta_k(\boldsymbol{\theta}_k) = \sigma_k(\boldsymbol{\theta}_k) - \rho_k$ representing the $k$-th step state difference between two density matrices; $\Gamma_k$ is the maximally mixed state $I/d$ where $I$ is the identity operator of dimension $d = 2^{w_k-1}$. $\Gamma_k = 1$ when $w_k = 1$. The bra-ket operation in the analytic form, $\langle A \rangle_\alpha = \langle \psi_{k-1} | U_k^\dagger(\boldsymbol{\theta}_k) A U_k(\boldsymbol{\theta}_k) | \psi_{k-1} \rangle$ for some Hermitian operator $A$ is evaluated at $\theta_k^\mu = \alpha$. This quantity of $G_k$ in equation 7 indicates the expectation value of $G_k$ regarding the $k$-th step variational ansatz

| | Global QNN | QSSM with max. layer widths | | | |
|---|---|---|---|---|---|
| | | 2 | 3 | 4 | 6 |
| **Physical States** | | | | | |
| XXX model GS | 0.533 | 0.523 | 0.883 | **0.915** | **0.956** |
| XXZ model GS | 0.523 | 0.750 | 0.887 | **0.954** | 0.952 |
| LiH molecule GS | 0.531 | **0.978** | **0.976** | 0.967 | **0.982** |
| **Algorithmic States** | | | | | |
| GHZ state | 0.535 | **0.994** | **0.993** | **0.992** | 0.978 |
| W state | 0.527 | **0.990** | **0.992** | 0.982 | 0.985 |
| Gaussian distribution | 0.561 | **0.969** | **0.985** | 0.976 | 0.986 |
| MNIST data encoding | 0.330 | 0.517 | 0.759 | 0.891 | **0.903** |
| Random state | 0.317 | 0.338 | 0.768 | 0.856 | 0.889 |

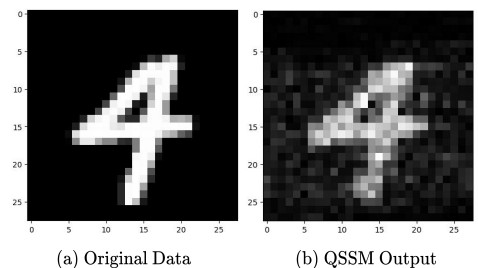

(a) Original Data          (b) QSSM Output

Fig 2: Effectiveness validation of QSSM in learning diverse 12-qubit quantum states regarding their final state fidelities. On the right, we show the QSSM learnt state (b) from the MINST dataset concerning the original data (a) using amplitude encoding. With different maximum layer widths, our QSSM outperforms global QNN on state learning tasks.

$|\psi_k\rangle$ evaluated at $(\theta_k^\mu)^* \pm \pi/2$ where all other scattering layers remain unchanged. The detailed derivation of these definitions and forms can be found in Appendix D.

Each partial derivative of $C_k$ at $\boldsymbol{\theta}_k^*$ can be explicitly determined by equation 7, which can be efficiently computable via shifting the corresponding parameter and applying variational quantum eigensolver Peruzzo et al. (2014). The gradient-based optimization could be applied to the cost by specifically updating the parameters $\boldsymbol{\theta}_k$ in the $k$-th layer as,

$$\boldsymbol{\theta}_k \leftarrow \boldsymbol{\theta}_k^* - \eta \nabla_{\boldsymbol{\theta}_k} C_k(\boldsymbol{\theta}_k^*) \tag{9}$$

where $\eta$ is the learning rate settled for the classical optimizers, defining the iteration step size. The cost function would converge to the optimal minimum by iterating the training processes. We then repeat the above procedures for each $k$-th layer to complete the model training with a final output circuit representation $\mathbf{U}(\boldsymbol{\theta}^{opt}) = U_n(\boldsymbol{\theta}_n^{opt}) \cdots U_1(\boldsymbol{\theta}_1^{opt})$ to finish the state learning.

## 5 NUMERICAL EXPERIMENTS

As described above, the adaptation of our quantum sequential scattering model indicates the underlying enhancement of information diffusion in quantum state learning. We now present numerical experiments to illustrate the effectiveness and trainability of QSSM.

We first conduct numerical simulations on QSSM for learning 12-qubit quantum states with physical or algorithmic meaning and compare our results with the performances from the conventional QNN model. The ground states from Heisenberg (XXX & XXZ) models Takahashi (1971) and the LiH molecular model are pre-determined via the OpenFermion library developed by McClean et al. (2020). For the Gaussian distribution and MNIST data learning experiments, the distribution and image data are normalized and mapped to the unit quantum state vectors of dimension $2^n$ via amplitude encoding Schuld (2021) with automatic padding of 0's filling out the extra grayscale pixels.

In our numerical simulations involving the global QNN and the QSSM, we employ a general hardware efficient ansatz (HEA) Kandala et al. (2017) of depth $d = 20$ with random initialized parameters for both the global model and each scattering layer in QSSM. The optimization uses the ADAM optimizer with a learning rate of $0.1$ and cost tolerance $0.001$, spanning 200 iterations.

As shown in Fig. 2, comparing the outcomes with those of the global QNN, we discern clear advantages exhibited by QSSM, which consistently attains notably high fidelity in learning diverse quantum states. Conversely, the conventional model does not perform well, primarily due to the significantly decreased convergence speed during the training processes with a large number of qubits.

Besides, states with exponential growth in Schmidt ranks are not necessarily hard to learn. Only highly entangled states, e.g., random states and maximally entangled states (MES) Gisin & Bechmann-Pasquinucci (1998), are challenging for QSSM. Those with concentrated Schmidt co-

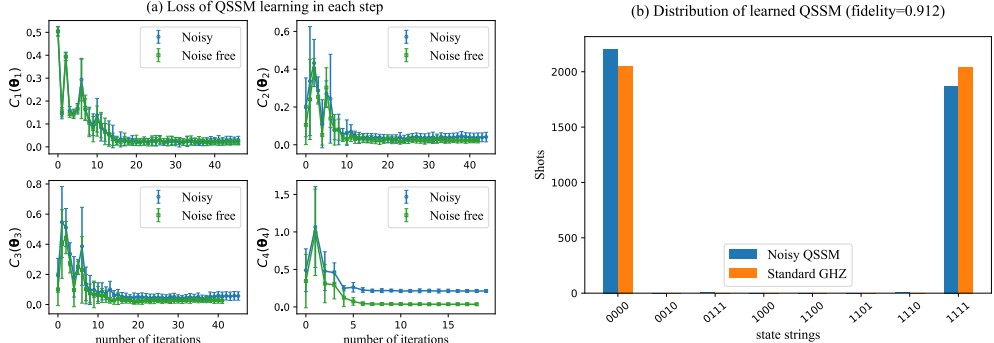

Fig 3: Noisy quantum simulation of QSSM for learning a 4-qubit GHZ state. (a) Comparison of the variation of cost function noisy quantum simulation and noise-free simulation. For both cases, the optimization was processed via COBYLA optimizer Gomez & Hennart (1994) on swap-test estimated cost values. (b) The distribution of measurement outcomes generated noise-freely from the state obtained by the noisy trained QSSM. The figure validates the efficacy and efficiency of QSSM in noisy environments, consequently reinforcing our method's practical applicability.

efficients, though owning large ranks, can be learnt up to a high fidelity Liu et al. (2022b) with limited resources.

In Table 2, we reasonably constrain the maximum scattering layer widths to some fixed values, which counterintuitively yield superior performance with smaller layer width. Larger values of $w_{\max}$, contrarily, decrease the QSSM performances of state learning. A plausible explanation for this phenomenon could be the over-parameterization and the mild BP effect during the training of the halved-dimensional scattering layers. Notably, learning random state undoubtedly obtains the worst learning results.

We also examine the noise robustness of using QSSM to learn a 4-qubit GHZ state on the IBMQ Qiskit simulator Qiskit contributors (2023). We build our noise model from single qubit and multi-qubit depolarizing channels (DCs) and thermal relaxation channels (TRCs) Georgopoulos et al. (2021). The error rate of DCs are set to $10^{-3}$, and the $T_1$, $T_2$ and gate time of TRCs are set to $1000 \, \mu$s, $100 \, \mu$s and $1$ ns respectively.

At each step, we run the optimization of the QSSM circuit 20 times in parallel and use the parameters that correspond to the lowest cost to update the circuit before going to the next step. This trick can significantly alleviate the randomness arising from sampling of bit strings in the measurement of quantum circuits. Shown in Fig. 3, each learning step has cost converged well compared with the ideal training in (3a). The final fidelity between the quantum state generated from QSSM and the true GHZ state could reach 91%, giving almost the same statistical behaviours plotted from the sampling experiments (3b).

From the analytical description and numerical demonstration, we see that QSSM has the ability to learn arbitrary quantum states with high fidelity compared to the conventional model. The diffusion strategy only requires narrow circuits in learning quantum states that are weakly entangled, thus being extremely efficient in learning such a class of quantum states.

We then present the result to demonstrate Proposition 2 by comparing the gradient variances of cost equation 3 as a function of the number of registers for QSSM and global QNN model. We typically investigate the values in the first step, the middle step ($\frac{n}{2}$-th step), and the last step of the QSSM learning procedure by looking into a single parameter $R_Z$ gate in the middle of each scattering layer. By assuming the two parts $U_{\pm}^{(k)}$ split by the $R_Z$ gate are deep enough to form local unitary 4-designs, we sample local Haar random unitaries Dankert et al. (2009) to simulate the behaviours of random initialization on $U_{\pm}^{(k)}$ and compute the gradient variances with respect to the parameter in $R_Z$. Similar experiments are performed for the conventional QNN model by sampling global Haar unitaries with a $R_Z$ gate sandwiched in. We target the GHZ state and the ground state

of the Heisenberg model, as before, with maximum width $w_{\max}$ being 2 and 4, respectively. The variance values are computed from sampling 500 Haar unitary pairs for both cases.

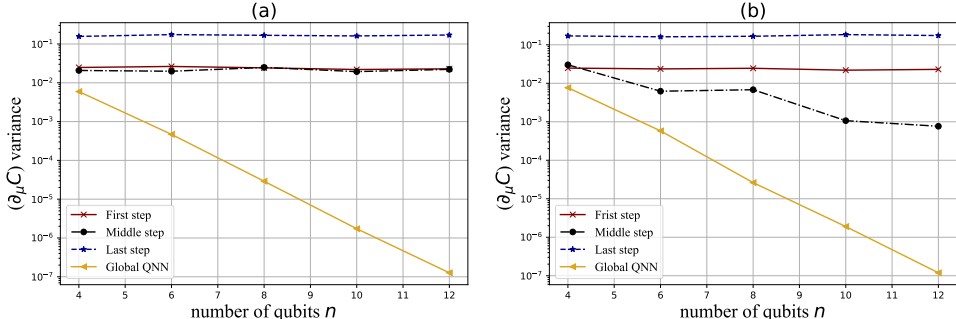

Fig 4: Comparison of the gradient variances as a function of the number of qubits on a semi-log plot from different steps in QSSM and global QNN computed by sampling Haar random unitaries. Panel (a) and (b) correspond to the learning of the GHZ state and the ground state of the Heisenberg model, respectively. The red, black and blue lines represent the gradient magnitudes of the first step, $\frac{n}{2}$-th step and the last step training, respectively, comparing with the global QNN results in yellow. Our method apparently outperforms conventional global QNN in terms of gradient variance scaling, indicating the absence of barren plateaus.

As we can observe in Fig. 4. The variance of the gradient vanishes exponentially with the number of qubits when using the randomly initialized global QNNs. In contrast, QSSM demonstrates a constant scaling of variance magnitude. We note that there is a decay of the gradient variance of the middle step in panel (b). Nevertheless, this decay is caused by a constant factor $g(\rho_k)$ that originates from the nature of the physical system and does not exponentially influence the training processes.

## 6    CONCLUSION AND DISCUSSION

In this paper, we have presented the development and application of the Quantum Sequential Scattering Model (QSSM) for quantum state learning. Our model is inspired by the classical diffusion model, which the designing of it involves quantum information theory and adaptive quantum machine learning techniques. Our theoretical analysis and numerical experiments demonstrate the superiority of the QSSM over conventional QNN approaches in terms of training speed and learning accuracy. In particular, the QSSM addresses the barren plateaus issues and provides an efficient solution to learning high-dimensional unknown quantum states based on sequentially learning the reduced target states.

Moreover, We have analyzed the impact of increasing entanglement, a key property of quantum states, on the performance and efficiency of the QSSM. Our results show that the model can effectively handle polynomially increased entanglement, enabling us to learn complex quantum states accurately. Numerical demonstrations have shown out-performances for learning physical and algorithmic quantum states in terms of their rank-restricted approximations, indicating the broad applicability of QSSM state learning and the deep connection between state learning and quantum entanglement.

There are remaining issues of QSSM for future discussion. Different choices of scattering layers would influence the learning performance, which has to be exemplified. How to further improve the state fidelity provided the high fidelity state from QSSM could become a significant open question. Understanding and resolving the effect of over-parameterization from QSSM should be explained. A theoretical performance guarantee and the connection between scattering layer dilation and QSSM state learning information flow should be established for a complete story of truncated state learning. We also expect some extended applications of QSSM as a new quantum generative model instead of only state learning on near-term quantum devices.

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
