# OpenReview forum: "Quantum sequential scattering model for quantum state learning"
_ICLR.cc/2024/Conference — ICLR 2024 Conference Withdrawn Submission_

### Official Review · Reviewer_y9Pp · 2023-10-31

**Soundness:** 3 good
**Presentation:** 2 fair
**Contribution:** 3 good
**Rating:** 5
**Confidence:** 3

**Summary:**

This study introduces a quantum sequential scattering model (QSSM) which is inspired by classical diffusion models, to address scalability issues and deal with the vanishing gradient problem for high-dimensional target states with polynomial-scaled Schmidt ranks. It gives the theoretical analysis and numerical experiments to support the effectiveness of QSSM.

**Strengths:**

The study developed a quantum learning model for state preparation with avoidance of barren plateaus and good training efficiency. From a theoretical and numerical perspective, it provides the evidence of effectiveness of the proposed model with proof-of-principle experiments.

**Weaknesses:**

1. the numeric experiments are insufficient, as it said that the proposed method has the advantages on training efficiency, but the numerics do not provide related comparison evidence.
2. whether the proposed learning model can handle quantum states with long-range interaction is not so clear.

**Questions:**

1. as discussed in algorithm 1, the proposed PQC only contains local operations, if the given state has long-range interactions, is there any guarantee that the proposed ansatz can also handle such states?
2. In the last paragraph of page 4, it said that the QSSM significantly reduces the parametric degrees. In the circuit model, what is the reduction scale of parameters of trainable gates compared to general QNN, such as the model with hardware efficient ansatz.
3. Is there any intuition guide for the selection of the max layer width?
4. In numerics, does it also apply the HEA as $U_k$ to the quantum register with d=20 for each scattering layer as the same as global QNN?
5. In fig2. the fidelity is a single run or average? if it is a single one, it is better to provide the average case.

---

### Official Review · Reviewer_6NCB · 2023-11-01

**Soundness:** 2 fair
**Presentation:** 2 fair
**Contribution:** 3 good
**Rating:** 5
**Confidence:** 2

**Summary:**

The paper proposes a new method, QSSM that improves quantum state learning. Specifically, the method tries to alleviate the problem of barren plateaus, by a hierarchical approach to sequentially learn reduced target states. This approach is inspired by classical diffusion models. The new algorithm is theoretically analyzed regarding its efficiency and indicating improved trainability which alleviates the barren plateau problem. Finally, the algorithm is empirically evaluated against a global QNN on diverse quantum state learning problems, as well as characterized under noisy conditions.

I am impressed by the asymptotic improvements of gradient variances which are also confirmed in experiment but this assessment is with low confidence. The main problem I see is that authors seem to ignore prior work aiming to improve QNNs already in both theory and experiments.

**Strengths:**

- The approach seems to be well motivated as a way to overcome the “barren plateau” phenomenon. While it does not circumvent it, it proposes to solve sub-problems that do not suffer so much as the full problem.

- The experimental analysis of the gradient variance convincingly shows the benefit of QSSM over global QNNs.

**Weaknesses:**

- The authors claim QNNs received “massive attention” in recent years. Works related to avoiding barren plateau are listed in the top paragraph of page 2. However, there is insufficient discussion about how these works relate to the proposed method.

- The choice of experiments would need further motivation, e.g. why are those tasks challenging for prior methods / useful in practice?

- IMO, the most severe drawback of the paper is that the authors only compare their method to the global QNN. Why did the authors not compare to any of the listed works that already tried/claimed to solve the Barren Plateau problem? If, for whatever reason, such comparisons are not applicable, they should at least explain why these comparisons do not apply. This critique applies to both theory and experiments.

**Questions:**

- How would "classical" ML algorithms perform on the considered task, i.e. generating rho'. (Might be a dumb question, as authors explicitly try to devise a better QNN learning algorithm.) Is there a fundamental computability limit that quantum computing approaches can overcome for such a task?

- Proposition 1 seems to be vacuous. Do I get that correctly that the correct state "could" be produced by QSSM if $U_k$ has a certain width w_k? So I don't know this width in practice and even if it is large enough, the algorithm may very well produce some $\sigma \neq \rho$? I know it is "advertised" only as a sufficient condition, but still this seems very weak to me. Still interesting though, is that they find that the necessary $w_k$ scales logarithmically.

- As far as I understood the Barren Plateau problem, the probability that the gradient vanishes grows exponentially in n, i.e., the variance of the gradient becomes exponentially small, i.e., O(2^⁻n). Authors claim their method achieves O(2^-n/2) and in some situations even O(1). This analysis only compares to the “naive” QNN and not to any of the other works trying to overcome the problem already?

- How were the experiments chosen? Are those widely considered benchmarks in the field or do they represent especially challenging tasks for the QNN approach? Also, what is the idea behind the (high dimensional?) Gaussian distribution and MNIST? What is the input there and what is the target?

- Why does the cost magnitude increase for each step (Figure 3 a)?

- Where is Table 2? Is it the one that is part of Figure 2? What are the numbers in this table (I think it is the fidelity), how are they obtained, what would be "worst" and "best" performance, what do bold numbers mean?

- Why are there no confidence intervals? Parameters are random initially, therefore not clear what re-runs of QSSM would produce.

- How would QNN compare on the MNIST task shown in Figure 2?

- How can it be that more width sometimes "hurts"?

- Is there any intuition, why the variances in the n/2-th step is lower than in the first step, but the last step is the highest of all three (Figure 4)? Also, how can the variance tend to decrease for the n/2-th step (Figure 4 b), but stay constant for the first and last step?

**Minor Remarks**:

- The relation of the approach to diffusion models that is stated throughout the paper could be explained more plainly.

- Enumeration of main results would be better in introduction

- Subsections / paragraph headers in experimental section would increase comprehensibility of experiments

- The GHZ state is not introduced in the main paper, only in supplements. Given that a major experiment utilizes this concept and an example for the usefulness of the variance bound on the gradient is based on it, it should be introduced properly in the main paper.

---

### Official Review · Reviewer_a2rC · 2023-11-05

**Soundness:** 1 poor
**Presentation:** 2 fair
**Contribution:** 1 poor
**Rating:** 1
**Confidence:** 4

**Summary:**

In the paper under review, the authors present a quantum neural network model, QSSM, which aims to integrate the concepts of quantum information diffusion with adaptive quantum state learning. The model boasts a new state learning algorithm that operates on a sequential subsystem-learning strategy, and the authors have theoretically demonstrated the algorithm’s effectiveness. They claim a polynomial-scaled gradient variance for the QSSM, suggesting a potential method to avoid barren plateaus in rank-restricted state learning.

The authors have carried out numerical experiments to validate their model, including the assessment of quantum states subject to noise effects, and compared their model to conventional QNNs in terms of training efficiency and learning accuracy. They advocate that their model exhibits enhanced performance on state learning tasks, supported by their comparative analysis.

Overall, the paper contributes to the discussion on quantum neural network architectures and their capability to learn quantum states effectively, adding to the broader conversation on quantum machine learning and algorithm development.

**Strengths:**

The paper introduces the QSSM model, a novel quantum neural network design that marries the concept of quantum information diffusion with adaptive quantum state learning, which could be seen as a thoughtful combination of existing theoretical frameworks.

**Weaknesses:**

While the paper attempts to contribute to the field of quantum machine learning with the QSSM model, there are critical issues that necessitate a strong rejection in its current form. The proposal of a polynomial representation of quantum states within classical architecture directly conflicts with widely accepted complexity theory. This discrepancy is not adequately reconciled in the paper and fundamentally undermines the proposed model's theoretical basis, as it deviates from the established understandings presented by Aharonov and Ta-Shma.

The benchmarking methodology is another significant concern. The choice of a QNN solver, known to be suboptimal for state learning tasks, as the primary point of comparison, lacks rigor and fails to convincingly demonstrate the superiority or novelty of the QSSM model. More appropriate and challenging benchmarks are essential for a fair assessment of the model's capabilities.

The treatment of purification schemes and the details of the Uk circuit are insufficiently addressed, missing the necessary clarity to be deemed innovative. This lack of detail leaves the claims of the paper unsubstantiated.

Additionally, the paper’s approach to mitigating barren plateaus by limiting parameters introduces a severe restriction on the model's representability, yet this critical trade-off is not thoroughly explored. The implications of such a design choice on the model's scalability and learning capacity are not adequately discussed.

Lastly, the paper's empirical results – notably the suboptimal fidelity in learning the GHZ state – are not only below the benchmark set by existing literature but also indicative of possible fundamental flaws in the proposed architecture or learning algorithm.

Due to these substantial shortcomings, which collectively cast doubt on the validity and contribution of the paper to the field, a strong rejection is recommended. Without a thorough revision that addresses the theoretical inconsistencies, expands upon the experimental comparisons, and offers a more in-depth analysis of the proposed architecture’s implications, the paper does not meet the standards for acceptance in a computer science conference.

**Questions:**

1. How does the proposed polynomial representation of quantum states align with the complexity theory limitations highlighted by Aharonov and Ta-Shma?
2. Can the authors provide a more rigorous benchmarking methodology that compares the QSSM model against more advanced and suitable QNN solvers?
3. In what ways does the QSSM model's approach to state purification differ from traditional methods that involve ancillary systems, and can the authors offer a more detailed explanation?
4. Could the authors elaborate on how the limitation to poly-logarithmic parameters affects the representability and learning capacity of the QSSM model?
5. How does the QSSM model account for the empirical results, particularly the lower fidelity in learning the GHZ state compared to existing literature?